# Virtual Agent Positioning Driven by Personal Characteristics

Anonymous

Submission ID: 2210

## ABSTRACT

When people use agent characters to travel through different spaces (such as virtual scenes and real scenes, or different game spaces), it is important to reasonably position the characters in the new scene according to their personal characteristics. In this paper, we propose a novel pipeline for relocating virtual agents in new scenarios based on their personal characteristics. We extract the characteristics of the characters (including figure, posture, social distance). Then a cost function is designed to evaluate the agent's position in the scene, which consists of a spatial term and an personalized term. Finally, a a Markov Chain Monte Carlo optimization method is applied to search for the optimized solution. The results generated by our approach are evaluated through extensive user study experiments, verifying the effectiveness of our approach compared with other alternative approaches.

## CCS CONCEPTS

• **Human-centered computing** → **Interaction design; Human computer interaction (HCI)**.

## KEYWORDS

Human-centered Computing, Scene Understanding, Virtual Agent Positioning.

### ACM Reference Format:

Anonymous. 2018. Virtual Agent Positioning Driven by Personal Characteristics. In *Proceedings of Make sure to enter the correct conference title from your rights confirmation emai (Conference acronym 'XX)*. ACM, New York, NY, USA, 9 pages. https://doi.org/XXXXXXX.XXXXXXX

## 1 INTRODUCTION

As an emerging trend in the digital age, virtual characters can flexibly travel through different spaces. People can perform activities from locations mapped from the real world to virtual reality. The same game character can also travel freely between different game spaces. The emergence of virtual agents has changed the traditional social model and provides more possibilities for people to meet in different spaces. However, it brings up a new question: when the character travels to a new space, where should he/she be positioned? Especially when there are multiple agents in the scene that need to be positioned, the positioning of the characters will be affected by each other. Properly positioning the agent in the scene

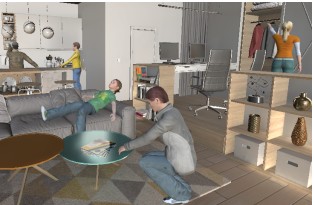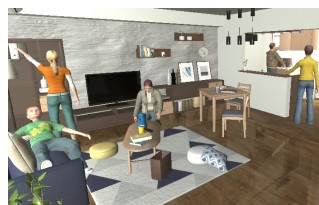

**Figure 1: Our approach learns the character's comfort zone in the scene through the user's position and posture in the original scene (left), thereby placing the virtual agent in an appropriate position in the new space to make the user comfortable and natural (right).**

greatly improves the user experience and allows the agent to better integrate into the new environment.

When people orient themselves in a scene based on a given pose, they first sift through recalling similar positions from previous experiences. Then they use their own experience to find reasonable objects in the new scene that can interact with the given pose to help determine the appropriate location, such as key interactive objects, special terrain. They finally obtain information from social interactions, observing other's positions and behaviors to find the appropriate position for their own interactive behavior. However, these individual-related experiences are difficult to generalize so that can quickly predict the optimal position of an agent representing an individual in a scene.

In recent years, some scholars have proposed approaches to reasonably locate agents in a scene[19, 38]. They locate agents in the scene based on the relevance of interaction behaviors, but do not consider the personalized characteristics of the agents themselves. The external characteristics, such as body shape and posture, will have an impact on their positioning in the scene. Agents cannot collide with other elements in the scene. Large one may sometimes have trouble standing in tight spaces. Intrinsic characteristics, such as personality and safe social distance, can also help determine the positioning of agents in a scene. For example, people tend to stand around people they are more familiar with and distance themselves from people they are unfamiliar with.

Based on the personalized characteristics of the agent, different agents have different probabilities of appearing in the same position of a scene. With the development of 3D technology, information in the scene can be easily extracted, such as scene scanning and reconstruction, virtual equipment, etc. The personality characteristics of an agent can be divided into personal attributes and relationships between individuals. Personal attributes include the character's body shape, posture, and the positioning relationship between multiple people can also reflect the social relationship between them. To better model the agent's personalized characteristics, we define a cost function to measure these feature relationships. Then we regard the positioning problem as an optimization problem and use

the Markov chain Monte Carlo optimization technology to perform personalized positioning of virtual agent.

In this paper, we propose a new pipeline for positioning agents in new scenes based on personal characteristics in historical scenes, which can help agents migrate in different scenes automatically. Through the application of this framework, we can position the characters in the virtual game in VR games based on the positions of multiple people in the same real scene, giving them a more realistic immersive experience. In addition, multiple users in different spaces can also be transferred to a new virtual scene together according to each character's position in the historical space.

The major contributions of our work are as follows:

- Introducing a new problem of personalized positioning of virtual agents based on character characteristics to freely travel through different spaces.
- Designing a computational framework to evaluate an agent's positioning in a scene, which combined agent characteristics and spatial constraints and then is applied to optimize the position of virtual agent.
- Conducting an extensive user study to evaluate and validate the effectiveness of the proposed virtual agent positioning approach.

## 2 RELATED WORK

In this section, we provide a succinct overview of the virtual agent positioning, and review the previous works in personalized characteristic modelling and social distance assessment.

### 2.1 Virtual Agent Positioning

The core of virtual agent positioning is to confirm the position and direction of the virtual agent, which is particularly important in the field of robotics, such as path planning [2, 11] and navigation systems[25]. These systems rely on algorithms, such as A* [10, 18] and Dijkstra[27, 40], to find optimal paths that enable virtual agents to move freely within the virtual environment. Ye et al.[41] present a position-aware virtual agent locomotion method, called PAVAL, that can perform virtual agent positioning in real time for room-scale VR navigation assistance.

To position the agent accurately, some studies control the behavior of virtual agent through predefined rules [32, 36], including avoiding obstacles[8, 14], finding cover, following scene rules, etc. While these methods have some feasibility, they are often not flexible enough to cope with diverse scenarios. With the development of reinforcement learning, some works attempt to train the best behavior strategy of virtual agents based on reward information in the environment[9, 23], which is more robust but also requires a large amount of training data and computing resources. Liang et al[19] also proposed a method to understand the scene with the help of scene semantics, allowing virtual agents to position themselves appropriately in the real world. Itsuki Noda[4] proposed an agent localization mechanism for dynamic environments using Delaunay triangulation to approximate the map. It can handle the positioning problem of multiple agents in the scene. However, these methods do not consider the personalized characteristics of virtual agents.

Compared with previous works, our approach mainly focuses on the problem of positioning multiple agents in the scene, taking into account the personal characteristics of each agent.

### 2.2 Personalized Characteristic Modelling

The comfort of the user experience is directly related to the fit of personal characteristics. With the upgrade of user experience, various fields have begun to work on personalized feature modeling, including interior design, education, human-computer interaction and so on. The most original method is to extract information directly from the communication of characters, such as face-to-face interviews, questionnaires, focus groups[1], and so on. These methods are time-consuming and their effectiveness depends entirely on the experience of the professional.

Some recent works extract personal habits through human daily life and activities in order to extract personality characteristics more realistically. Wang et al[34] constructed action graphs by observing videos of daily activities, which were used to train a generative model based on recurrent neural networks (RNN). This method models personal characteristics accurately, but requires significant storage costs. Some researchers propose that personal characteristics can be learned from virtual environments by simulating real activities. Wei et al.[21] proposed to a pipeline to learn personal preferences from virtual experiences, which was used in indoor furniture layout. Other studies look for clues of personal characteristics from historical scenes. For example, Wang proposed to use the previous home layout to layout furniture in a new scene.

Inspired by these studies, we propose to extract the personality characteristics of the characters in the previous scene, which are then used in the personalized positioning optimization of the characters in the new scene.

### 2.3 Social Distance Assessment

A safe social distance is a certain distance that people should maintain between themselves and others in different social and environmental situations to ensure physical health and social comfort[6, 22]. It plays an important role in social interactions in daily life. If two people stand too close together in a scene, each other will feel oppressed and tense. If the two people stand too far apart in the scene, it will affect communication and interaction with each other[30].

Traditional methods typically use mathematical distance measures, such as Euclidean distance[3] or Manhattan distance[29], to measure the distance between people. For example, with the outbreak of the Corona virus in the past few years, more and more public areas have begun to pay attention to the safe distance between people, and have embedded cameras to detect whether the social distance between people reaches 1.2m[15]. These methods provide a basic way to assess social distance, but ignore the factors of individual differences and social comfort. Some studies have attempted to assess social distance through behavioral modeling. These methods take into account interactions and behaviors between individuals, such as posture[20], facial expressions[5], and how they move. Gao[20] conducted an experiment to investigate the effect of posture and embodiment on the social distance of participants and agents during interactive experiences in mixed reality. Leon[17] established personalized models of social safety distance by collecting users'subjective feelings and feedback in different scenarios, which may require large amounts of user participation and feedback data and are difficult to apply in real time.

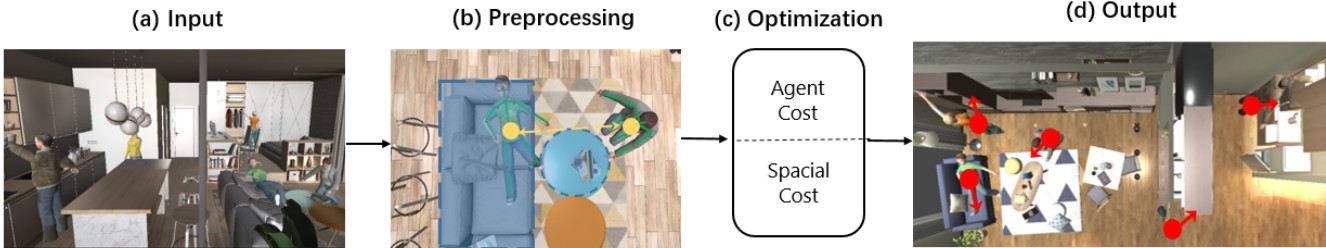

**Figure 2: Overview. Given the postures and positions of multiple people in the original scene (a), we model the personal characteristics and social comfort zone of each agent(b). Through a Markov chain Monte Carlo optimization technique (c) We reposition the virtual agent in the scene (d).**

Different from the previous works, we model an individual's comfortable social zone through the agent's position in the virtual environment, which makes social distance assessments more personalized.

## 3 OVERVIEW

Given a historical scene containing agents as input, our approach aims to optimize the agent's positioning in the new scene so that the agent can stand in a more reasonable position to perform tasks when transferred across scenes. It should be noted that the historical scene and the posture and information of agents can be obtained based on three-dimensional reconstruction[37, 42], or can be directly captured based on the RGB camera of the VR device[39], such as HoloLens.

Inspired by the method of finding appropriate positioning in a scene, we designed a framework to automatically position multiple agents in new scenes based on personal characteristics in historical scenes. When a person tries to find a suitable position in a new scene, he must first call upon his previous memories to perceive the environment (what objects he is interacting with and how others around him relate to himself), and then find a position and direction in the scene that is suitable for the current action and task. Therefore, the first step of our approach is to model the environment (character actions, interactive objects, and relationships between characters in the scene) based on historical scenes. In this paper, we propose a method for encoding environments from historical scenes and then optimizing the position of each agent in new scenario through heuristic search.

Assume that there are $N$ virtual agents in the scene, and their layout positioning in the scene is defined as $L = \{l_i | l_i = (x_i, y_i, z_i, \theta_i), i \in \{1, 2, ...N\}$. Specifically, for the $i$th individual, $(x_i, y_i, z_i)$ is the position of the agent, and $\theta_i$ is the orientation of the agent. Based on the layout $L_0$ of characters in historical scenes, our goal is to automatically generate reasonable character layouts in new scenes. The framework is shown in Fig. 2. Our approach includes two stages: preprocessing and optimization.

*Preprocessing.* Using image processing algorithms, we preprocessed the information of individuals in historical environments, including their positions, postures, relationships with objects, and relationships with other individuals. Firstly, we employed the Mask R-CNN method [13] to detect agents and key objects in the scenes. Then we further utilized OpenPose[28, 33] to extract the poses of

individuals, and employed Euclidean distance calculation to determine the distances between individuals as well as the angles between their directions. After these two steps of preprocessing, we modeled the effective information of the whole environment in the historical environment. We will discuss the details in Sec. 4.

*Optimization.* The optimization stage considers the characteristics of the characters in the environment and optimizes the position and orientation of the virtual agent iteratively. The cost function with personalized constraints and spatial constraints is designed to evaluate how well the position and orientation of each virtual agent is. The personalized constraints ensure that the character's position and orientation maintain a reasonable social distance and relative relationship with others. The spatial constraints control the reasonable position of the virtual agent's posture in the scene, which is to achieve smooth interaction of characters in the scene. A Markov chain Monte Carlo optimization algorithm is applied to search for solutions, which is discussed in Sec. 5.

## 4 PREPROCESSING

Following previous work, 3D scenes can be reconstructed through nerf[24], 3D gaussion splatting[16], differentiable rendering[26] and other method. To model the distribution of agents in a scene, it is necessary to extract the properties of the agents and the semantics of their interaction with the scene.

### 4.1 Agent-centric Extraction

In a given scene, the individual attributes of agents encompass their position, posture, as well as personal information such as height and body type.

*Position.* The agent's position is where the user is standing. Therefore, we use the midpoint of the user's two foot coordinate points as the user's position. During motion capture, the positions of each agent are written sequentially as $Loc = \{loc_1, loc_2, \cdots, loc_N\}$, where $loc_i$ is the set of 3D coordinate of the agent $i$ in the original scene.

*Posture.* The user's posture reflects the points at which the user needs support. We divide the postures according to the main supporting surface into standing, sitting, lying and leaning. To recognize the user's posture offline, we collected 1000 pictures containing various postures, about 250 pictures of each category, and trained

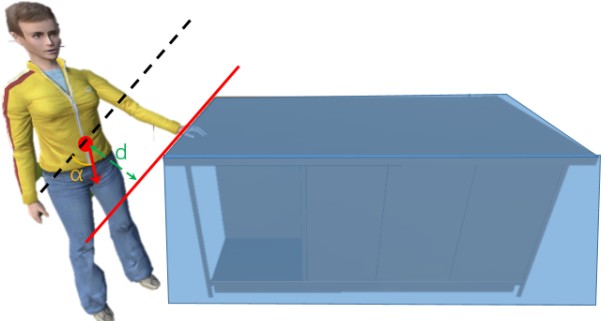

**Figure 3: An example of key object. The red dots and arrows indicate the agent's position and orientation. The blue box represents the key objects. The red solid line is the key edge closest to the agent from the key object. The green line is the vertical distance from the agent to the key object. The black dotted line is parallel to the key edge. The yellow angle indicates the relative direction of the agent and the key object, which is between the black line and the direction of the agent.**

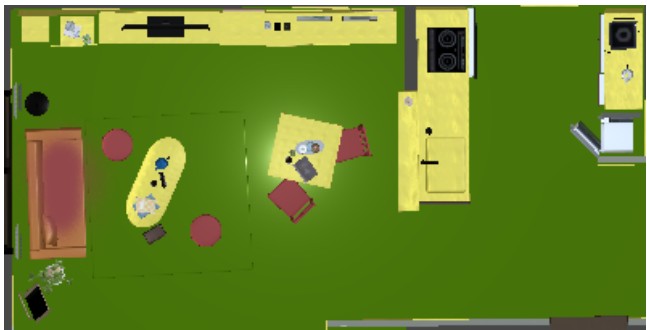

**Figure 4: An example of a plane in a scene that represents a supportable posture. Different colors represent supported movements, among which green represents the area where you can stand, red represents the area where you can sit, orange represents the area where you can lie down, and yellow represents the area where you can lean. It should be noted that some components have multiple colors, indicating that they can support multiple postures. e.g. sofa.**

an posture classification network based on VGG-16[35], with an accuracy of 98%. Each posture is coded as 1, 2, 3, 4 respectively. The posture of each agent is recorded as $Pos = \{pos_1, pos_2, \cdots, pos_N\}$ where $pos_i \in \{1, 2, 3, 4\}$, and $pos_i$ is the posture type of the agent $i$.

*Body size.* The body size reflects the space the agent needs in the scene. We use the IK algorithm to calculate and record the sequence of 16 key points(shoulders, elbows, torso, etc.) on the agent's body, with the foot posture as the origin of the coordinates. The Inverse Kinematics algorithm infers the positions of the other joints of the user by the Jacobian inverse technique in real time[7]. The body size of agent $i$ is defined as $J_i = \{j_1, j_2...j_{16}\}$, $i \in \{1, 2, ...N\}$. $j_i = (\theta_p, \theta_y, \theta_r, dis)$, where $\theta_p, \theta_y, \theta_r$ is the vector direction from the joint point to the origin, representing pitch, yaw, and roll of the joint and $dis$ is the distance from the joint point to the origin.

It is worth noting that through the recording of the body size, we can calculate the 3D coordinates of each key point for each agent in the scene according to the position of them. The coordinates of the $j - th$ key point of agent $i$ are recorded as $J_{i_j}$, where $i \in \{1, 2, ...N\}$ and $j \in \{1, 2, ...16\}$ .

### 4.2 Scene-centric Extraction

It is crucial for the agent's positioning to complete the interaction between the character and the key objects in a reasonable position in the scene using corresponding postures.

*Key Object.* Consider the closest object to each agent as the key object related to the agent. To make the environment in which the agent acts consistent in the new scene, the relative positional relationship between the agent and key objects should remain unchanged. As shown in Figure 3, taking the nearest neighbor edge of the key object as the target, we calculate the vertical distance from the agent to the target and the relative angle between the agent and it of each agent. We denote the vertical distance of agent $i$ as $d_i$ and the angle as $\alpha_i$.

*Plane.* Different planes in the scene can support different postures. We use the Mask RCNN algorithm [13] to detect the plane of each object in the scene. Figure 4 shows an example of a plane that can support various postures in a logo scene, where green is the place where you can stand, red is where you can sit, blue is where you can lie down, and yellow is where you can lean.

## 5 OPTIMIZATION

We randomly generate a set of localizations $L_0$ for each virtual agent in the new space. The problem of agent localization in new scenarios is then considered as an optimization problem, under personal and spatial constraints. In this section, we discuss the definition of the cost function and the optimization process in detail.

### 5.1 Cost

During the optimization, we tried to enable the agent to be placed comfortably in the new scene by analyzing the relative relationship between the virtual agent and the objects in the new space, the social relationship between multiple agents, and the personal characteristics of the agent. We define the cost function considering personality constraints and spatial constraints as follows:

$$C_{\text{total}}(L_{in}, L) = \omega_a C_a(L_{in}, L) + \omega_s C_s(L). \tag{1}$$

$C_a(L_{in}, L)$ is to assess the cost associated with personal characteristics extracted from the original scene. It penalize solutions that deviate from the expected interaction dynamics between the agent and scene elements, as well as the spatial distribution among all agents. $C_s(L)$ is to consider whether the agent is on a reasonable position in the scene, which is evaluated from the agent's admissibility and the distance between each agent. The $\omega$ coefficients determine the relative weighting of each item; in practice, we set $\omega_a = 0.5$ and $\omega_s = 0.5$.

*5.1.1 Agent Cost.* In personalized cost, we consider the relative relationships between human and scene, and the distributed relationships within agents. It is defined as follows:

$$C_a(L_{in}, L) = \omega_r C_r(L_{in}, L) + \omega_d C_d(L_{in}, L), \qquad (2)$$

where $C_r(L_{in}, L)$ is the interaction cost, and $C_d(L_{in}, L)$ is the distribution cost within agents. $\omega_r$ is set 0.6 and $\omega_d$ is set 0.4.

*Interaction.* When interacting in a new space, the relative relationship between the agent and the components of the scene with which it is interacting should remain as unchanged as possible. Once changed, it may affect the intent of the agent's posture. For example, it would make sense to raise a hand to try to grab something when the user is near a cupboard on the wall. Therefore, we consider the relative position of the user and key objects in the interaction cost. The cost is defined as follows:

$$C_r(L_{in}, L) = \frac{1}{N} \sum_i \left( \frac{|d_i - d_i'|}{2 \cdot d_{max}} + \frac{|\alpha_i - \alpha_i'|}{2 \cdot 90} \right), \qquad (3)$$

where $d_i$ is the distance of the agent $i$ from the edge of the nearest interactive object. $d_i'$ is the distance from agent $i$ in the new space to the edge of the nearest object in the space. $d_{max}$ is the maximum distance from all agents in the initial scene to key objects. $\alpha_i$ is the angle of the agent's orientation relative to the key edge. $\alpha_i'$ is the angle of the agent's orientation in the new space relative to its key edge. It should be noted that $\alpha_i$ and $\alpha_i'$ are constrained to 0-90 degrees. $N$ is the number of agents in the scene.

*Distribution.* The new space is different in area from the original scene. The density distribution of agents in the scene is proportional to the scene area. In order to distribute the agents reasonably in the new space, we first use the mean of K nearest neighbor distances to calculate the density distribution of the agents in the original scene, where K is 3. For each agent, we find the three closest agents in the scene and calculate the Euclidean distance between it and these agents. Then we calculate the average of the three distances as the density of agent. We reposition the agent in the new space based on the area of the space. The cost is defined as:

$$C_d(L_{in}, L) = \frac{1}{N} \sum_i \left| \frac{K_i - K_i'}{l} \right|, \qquad (4)$$

where $K_i$ is the social density of the $i$-th agent with other agents in the scene. $K_i'$ is the social density of the $i$-th agent with other agents in the new scene. $l$ is the value of the diagonal length in the scene. $N$ is the number of agents in the scene.

*5.1.2 Spacial Cost.* In the spacial cost, we consider the admissibility of people in the scene and the reasonable social distance around agents. It is defined as follows:

$$C_s(L) = \omega_{ad} C_{ad}(L) + \omega_{sd} C_{sd}(L), \qquad (5)$$

where $C_{ad}(L)$ is the admissibility cost. $C_{sd}(L)$ is the social distance cost. The $\omega$ coefficients determine the relative weight of each item; in practice, we set $\omega_{ad} = 0.5$ and $\omega_{sd} = 0.5$.

*Admissibility.* The positioning range of characters in the scene will vary depending on the character's posture and body size. For example, overweight people cannot complete some movements in some small places. The admissibility of an agent is mainly judged in two aspects: We detect the position of each key point of each agent in the scene to determine whether the positioning is feasible. In addition, it is also necessary to determine whether the user's posture is on a plane that matches the positioning. The cost is defined as:

$$C_{ad}(L) = \frac{1}{N} \max(0, \sum_i V(pos_i, loc_i), \sum_i V(J_i)), \qquad (6)$$

where $loc_i$ is the position of agent $i$. $pos_i$ is the posture type of agent $i$. $V(pos_i, loc_i)$ is to determine whether the user's posture $pos_i$ satisfies the scene plane label value at the location $loc_i$. If it is satisfied, $V(pos_i, loc_i)$ is 0, if it is not satisfied, it is 1, which can be calculated based on the label map of the scene. $J_i$ is the position of each joint point of agent $i$ in the scene. $V(J_i)$ is to determine whether the agent collides with an object in the scene. The value is 1 if the node in question is in an inappropriate position in the scene, 0 if not.

*Social distance.* Social safe distance refers to the distance maintained between people in social situations to ensure that each other feels comfortable and safe. Positioning too close can cause agents to feel uncomfortable or even cause collisions between agents. In reality, the safe social distance should not be less than 1.2 meters[31]. The same as the virtual environment, where there should be reasonable social distance between agents in a multi-agent layout. The cost is defined as:

$$C_{sd}(L) = \frac{1}{2N} \sum_i \sum_j \max(0, \frac{1.2}{D(loc_i, loc_j)} - 1), \qquad (7)$$

where $loc_i$ is the position of agent $i$ in the scene, and $loc_j$ is the position of agent $j$ in the scene. $D(loc_i, loc_j)$ is the Euclidean distance between agent $i$ and $j$. $N$ is the number of agents in the scene.

## 5.2 Simulated Annealing

To steer clear of getting trapped in local minima within the solution space, we employ a simulated annealing algorithm to systematically probe the arrangement of the virtual agent. This algorithm is adept at embracing suboptimal choices according to the Metropolis criterion. To elaborate, starting with the initial placement of the agent denoted as $L_0$ within a fresh spatial context, we proceed by selecting a potential relocation and then assess its viability using the Metropolis-Hastings acceptance rule. This acceptance rule is delineated as follows:

$$A(L^i, L^*) = \min\left\{1, \frac{p(L^*)}{p(L^i)}\right\}, \qquad (8)$$

$p(\cdot)$ is computed using by the defined cost:

$$p(\cdot) = \frac{1}{Z} \exp\frac{-C_{total}(\cdot)}{T}, \qquad (9)$$

where $C_{total}(\cdot)$ is defined in Section 5.1. $T$ represents the temperature of the annealing process. Initially, $T$ assumes a large value, enabling the sampler to traverse the solution space with heightened.

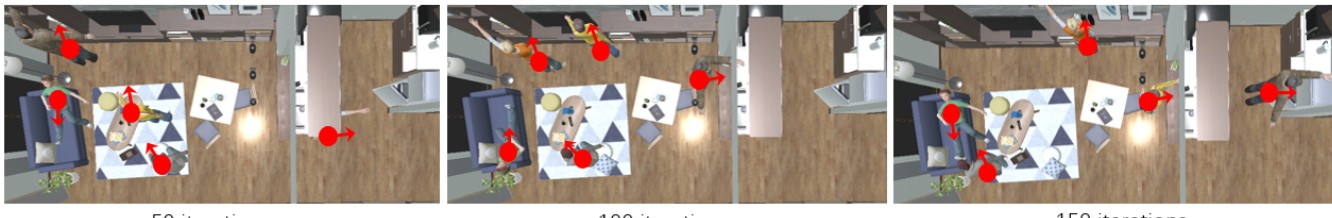

50 iterations         100 iterations         150 iterations

**Figure 5: An example of the optimization. The curve represents the cost change during the optimization. We also visualize three intermediate configurations on the bottom.**

Subsequently, as optimization progresses, $T$ undergoes gradual reduction. As it nears the conclusion, $T$ converges a small value near zero, facilitating finer adjustments to the solution. $Z$ denotes a normalization constant. By default, we empirically set $T$ to 1.0 and decrease it by 0.05 every 10 iterations until it reaches zero. Termination of the optimization is warranted if the absolute change in the total cost value registers below 5% over the past 20 iterations. In our experiments, a complete optimization spans around $140 \sim 300$ iterations.

To speed up the optimization, three strategies is proposed to do an action with a respective probability, which modifies the position and orientation of each agent:

*Translation.* Translation is the basic operation for changing the agent location in the scenario. Agent $i$ is selected and updated with the move. In the $(x, y)$-space position move, a translation move generates a new position $(x_k + \Delta x, y_k + \Delta y)$ based on the current location $(x_k, y_k)$. We sample the variation $(\Delta x, \Delta y)$ using a Gaussian distribution.

In the $z$-dimension, we set it in the new scene based on the height of the terrain of the $(x, y)$ plane. When there is no object at its location, that is, the agent is standing on the ground, $z$ is 0. Otherwise, the $z$ value is the height of the object in the corresponding position. For example, when the agent is placed on a bed, the $z$ value is the height of the bed.

*Rotation.* Rotation changes the orientation of the agent. In the rotation move, the sampler generates a new orientation $(\theta_i + \Delta\theta)$ based on the current orientation $\theta_i$. The rotation change $\Delta\theta$ is generated from the Gaussian distribution.

*Swapping.* Swapping refers to exchange the positions of any two agents in the scene, which can accelerate the exploration of the solution space. We choose two agents $i$ and $j$ at random and swap their positions directly.

It should be noted that we have enabled collision detection in the scene. If the agent and the scene collide through the model, the action is rolled back and the next action is selected.

## 6 EXPERIMENTS

In this section, we discuss several objective and subjective experiments conducted to evaluate the effectiveness of our pose synthesis approach. We implemented our approach using using C# and Unity 2021 and ran the optimization approach on a PC equipped with 32GB of RAM, a Nvidia Titan X graphics card with 12GB of memory, and a 2.60GHz Intel i7-5820K processor.

### 6.1 Compared approaches

To verify the proposed approach, we compared three approaches for virtual agent positioning:

- Multiple agents are put into new scenes in turn through the POSA[12].
- Professionals put several virtual agents into new scenarios in sequence based on their professional experience. We recruited three professionals who have been engaged in animation production for 5 years.
- Our approach automatically positioned the virtual agents in the new scene based on the character's characteristics of the virtual agents in the original scene.

We compared results of these approaches in quantitative and qualitative experiments.

*Validation Dataset.* The validation dataset consists of 25 scenes, both indoors and outdoors. We use the above three methods to place several agents for each scene. The number of agents in each scene is set to 5-10 depending on the size of the scene. First, we invite 5-10 people in each scene to simultaneously perform simulated actions in a scene similar to the operation of a simple social mobile game. The virtual agent of the new scene is then populated based on the scene positioning of the character in the original scene in our approach as well as that of the professionals. Compared with the method where professionals observe the scene, our method automatically analyzes the characteristics of the agent.

### 6.2 Qualitative Experiment

we carried out user studies to evaluate the effectiveness of our approach and the aesthetic experience subjectively.

*Participants.* We recruited 30 participants, with a diversity of backgrounds in terms of many aspects. The participants included 14 males and 16 females whose ages ranged from 12 to 55. Occupationally they range from unemployed or retired people to students, educators and business people. In addition, all the subjects reported normal or corrected-to-normal vision with no color-blindness.

*Procedure.* The 25 scenes were randomly divided into 5 groups, with 5 scenes in each group. 30 participants were also randomly divided into 5 groups, with 6 people in each group. Each group of participants was randomly assigned to a set of scenarios to rate multiple agent positions generated by the three methods. Specifically, the score mainly consists of two parts: i) the rationality of a

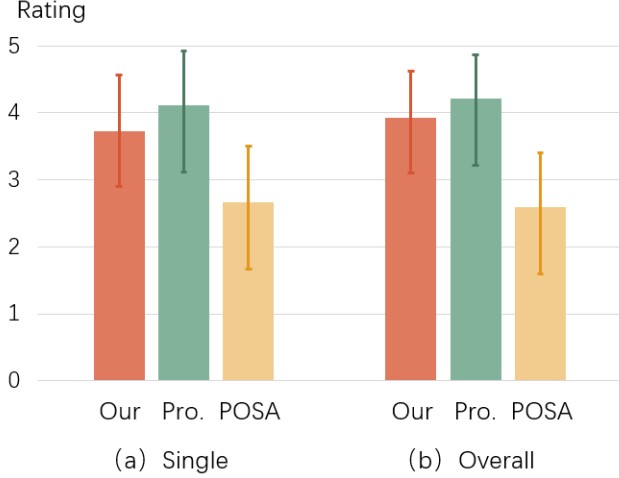

**Figure 6: Average user ratings on rationality of single-person postures and overall distribution of multi-person positions of different approaches (i.e. our approach, Professional approach, and POSA) in subjective evaluation.**

single character's position (to verify the rationality of each character's interaction with the scene) ii) the rationality of the overall character's position (to verify the rationality of the social distribution of all characters ). Ratings range from 1 to 5 on a 1-5 Liker scale, with 1 meaning inappropriate positioning and 5 meaning the opposite. Results generated by the three methods were presented to participants in random order to avoid bias. Not only that, we also conducted semi-structured interviews about user experience to further explore the factors that influence ratings. It should be noted that each set of scenarios is guaranteed to correspond to a set of participants.

*Results and Analysis.* The average score statistics of the 25 scenes on the rationality of single-person postures and overall distribution of multi-person positions are shown in Figure 7. From the statistical point of view, the positioning of professionals received the highest score in the two evaluations (single: $M = 4.10$, $SD = 0.76$; overall: $M = 4.17$, $SD = 0.65$), followed by Our approach (single: $M = 3.83$, $SD = 0.75$; overall: $M = 3.89$, $SD = 0.71$) and POSA (single: $M = 2.66$, $SD = 0.84$ ; Overall: $M = 2.58$, $SD = 0.81$). In all cases, our positioning approach are preferable to POSA and comparable to the Professional approach.

To verify the validity of our results, we also performed a OneWay ANOVA test on the mean ratings of each scenario using a $\alpha = 0.05$ significance level. The results show that there is no significant difference between our method and the manual method of professionals pose (single: ($F_{[1,49]} = 3.21$, $p = 0.07 > .05$), overall: ($F_{[1,49]} = 2.25$, $p = 0.134 > .05$). However, compared with the POSA approach, both approaches achieved significantly higher scores than POSA: the Professional approach and POSA (single:($F_{[1,49]} = 24.01$, $p < .05$),overall:($F_{[1,49]} = 23.15$, $p < .05$)), Our and POSA (single:($F_{[1,49]} = 25.01$, $p < .05$),overall:($F_{[1,49]} = 21.16$, $p < .05$)) It is proven that our approach can quickly learn character characteristics based on existing scenes and fill in new scenes, and the effect is comparable to that of professionals. This enhances the agent's

scene switching experience and allows empty scenes to be quickly and appropriately filled.

In our approach, the number of components in the original scene is 6. It is interesting that our method works better at positioning the agent in the new scene when it has the same number of components as the original scene.

*User Feedback.* By asking users for their opinions, we learned that the suitability of the user's clothing and the environment will also affect the user's overall perception of the scene to a certain extent. For example, a user wearing a chef's uniform may feel strange if he or she is standing in a non-kitchen area. Some users said that virtual agents look a bit monotonous because their expressions are uniform. Because in daily interactions, characters may have a variety of expressions. These feedbacks have provided us with some interesting improvements, helping us to consider users' personalized information, such as clothing, expressions, etc., in the scene auto-fill problem.

## 6.3 Quantitative Experiment

We verified the effectiveness of our approach quantitatively by comparing the number of collision points between the character's posture and the surrounding environment in each scene and the synthesis time of each approach.

*Collision Points.* The reasonable posture and position of each virtual agent in the scene should not cause any unreasonable cross-modeling with any model in the scene. In each scene, we calculated the number of points where cross-mode collisions occurred between the virtual agent and the scene in each method generation scheme by turning on collision detection in the scene. Among a total of 202 characters in all scenes, our method did not cause improper cross-modeling between the characters and the scene. The professional method had 3 collision points with the scene, while POSA had 89 collision points with the scene. The results show that our approach has obvious advantages due to the collision loss in our approach. Compared with manual work by professionals, there are always small mold-piercing errors caused by carelessness.

*Synthesis time.* For each scene, we recorded the synthesis time for each method of automatically populating the new scene. Specifically, the time records for each method are as follows:

- POSA: Place characters with fixed poses into the scene in sequence, starting from the time when the first character is placed until all characters are placed.
- Professional: Professionals put characters with fixed poses into appropriate positions in the scene, but professionals can make fine adjustments to the characters' poses to fit the scene. The timing starts from when he sees all the characters until all characters are adjusted.
- Ours: Our approach starts timing by extracting character relationships from the original scene until the repositioning and pose adjustment of all characters are completed in the new scene.

The results show that POSA's method uses the least time to fill the scene ($M = 0.3s$, $SD = 0.02s$). Our method fills the scene much faster ($M = 0.28s$, $SD = 0.03s$) compared to the professional approach ($M = 12.62min$, $SD = 1.71min$). By asking professionals, we

found that they would first spend 3 minutes viewing and analyzing the scene structure and each user's posture, then spend 1 minute initializing their position in the scene, and finally fine-tuning the character's posture according to the size of the scene. , to ensure the rationality of each pose in the scene. This process usually takes a long time, usually 8 minutes. The results shows that the positioning time of our method can realize real-time applications, such as rapid switching of multiple scenes during real live broadcasts or virtual games.

## 7 CONCLUSION

In this paper, we propose a new problem of how to personalize characters based on their characteristics. We attempt to optimize the problem of agent positioning in new scenes based on character characteristics learned from historical scenes. To achieve this goal, we attempt to model human characteristics from information in historical scenes, which are then used in the optimization of agent positioning in new scenes. Specifically, the external characteristics (such as figure, posture) and internal characteristics (social distance) of the characters are first extracted from the historical environment.Then a cost function is designed to evaluate the reasonableness of the agent's position in the scene. Finally, a heuristic algorithm based on Monte Carlo criteria is applied to optimize the location of the agent in the scene, which make it more consistent with individual characteristics.

Our approach leads to a variety of potential applications in both virtual and real worlds. For example, players often encounter scenarios where multiple characters need to seamlessly transition from one level or scene to another in virtual game development. By leveraging our approach, game developers can ensure that characters' positioning and interactions are consistent, enhancing the overall gaming experience and immersion for players. Another potential application area is in urban planning and architecture. For example, urban planners and architects can use virtual simulations of the distribution of crowds in a scene to visualize and analyze urban environments and building designs. This can help simulate human interaction and behavior in these environments, thereby informing more realistic and effective urban planning and building design.

*Limitation and Future Work.* The analysis of specific inputs may be inaccurate due to the focus on human subjects in scene annotation data. For example, inaccuracies can occur when the proxy object is an animal such as a kitten. Various cartoon characters and avatars are becoming more and more popular in virtual applications, especially in games. Future work should involve collecting and integrating behavioral data from multiple organisms to improve the accuracy of agent localization.

As characters transition from one scene to another, their attire should seamlessly complement the setting. For instance, when moving from a modern to an ancient-themed scene, it would be interesting if the character's wardrobe dynamically adjusts to align with the scene's style. Implementing such clothing adaptation not only ensures the logical positioning of characters during automatic scene transitions but also enhances their visual coherence with the new environment.

While our current approach focuses on modeling the static posture of characters, a promising future direction involves seamlessly integrating dynamic user interactions into virtual scenes. This advancement could be achieved by swiftly mapping the real-time actions of users, captured through wearable MR devices like Microsoft HoloLens, into a dimensional space within the virtual environment. By accurately translating the user's dynamic action sequences into corresponding positions within the virtual scene, we can propel the evolution of AR animation, enhancing immersion and interactivity for users.

## ACKNOWLEDGMENTS

To Robert, for the bagels and explaining CMYK and color spaces.

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

Received 20 February 2007; revised 12 March 2009; accepted 5 June 2009
