# OpenReview forum: "Virtual Agent Positioning Driven by Personal Characteristics"
_acmmm.org/ACMMM/2024/Conference — MM2024 Oral_

### Official Review · Reviewer_SJiL · 2024-05-16

**Rating:** 6
**Confidence:** 3

**Summary:**

The paper presents a novel approach for relocating virtual agents in new scenarios based on their characteristics. It outlines a pipeline involving the extraction of character traits, the design of a cost function evaluating the agent’s position, and applying a Markov Chain Monte Carlo optimization method. User study experiments confirm the effectiveness of the approach compared to alternatives.  The paper is accepted for publication pending minor typographical revisions.

**Strengths:**

The paper introduces an innovative methodology for optimizing agent positioning based on character characteristics in virtual environments. It addresses a pertinent problem in virtual scene navigation and offers a systematic solution. The proposed pipeline, comprising trait extraction, cost function design, and optimization, demonstrates a comprehensive understanding of the problem domain.
The writing style of the article is commendable, exhibiting clarity and coherence throughout. The use of language is precise, making complex concepts accessible to readers from diverse disciplines. The paper seamlessly integrates insights from various fields, highlighting its multidisciplinary nature.
The research contributes significantly to both theoretical and practical domains. By incorporating personal characteristics into agent positioning, the approach enhances realism and immersion in virtual environments. The potential applications, including virtual game development and urban planning, underscore the versatility and relevance of the proposed methodology.
Furthermore, the paper is well-structured, with each section logically progressing from problem formulation to solution implementation. The clarity of the presentation facilitates understanding and ensures a smooth reading experience for the audience.
In conclusion, the article offers valuable insights into personalized character positioning in virtual environments. Its rigorous methodology, clear exposition, and multidisciplinary relevance make it a valuable contribution to the scientific community. The paper is accepted for publication without revisions, except for two minor typographical errors and the suggestion to add some comment about the choice of the weights in equation 1 and 2. One of the typo mistake occurs in Section 6.2, "Results and Analysis," where "Our" is capitalized instead of lowercase as elsewhere in the manuscript. A missing space before "Then" in the conclusion has also been noted. These errors should be corrected before final publication.

**Limitations:**

One of the limitations highlighted in this scientific article concerns the choice of weights used in equations 1 and 2. It is emphasized the need to explore different combinations of weights to assess the impact on the performance of the proposed method. While choosing a weight of 0.5 may be reasonable, it would be desirable to conduct a more thorough analysis to understand how variations in weights affect the results. In particular, the article could benefit from a more robust justification regarding the selection of weights in equation 2. Exploring different combinations of weights could provide a better understanding of the model and allow for better optimization of the overall system performance.

**Suitability:**

2

---

### Official Review · Reviewer_8jV3 · 2024-05-24

**Rating:** 2
**Confidence:** 2

**Summary:**

The paper introduces a pipeline designed to position virtual agents in scenarios that adapt to their specific characteristics. By utilizing the position and posture of an agent in an initial scene, the method determines the most suitable placement for the same agent in a new scene. The primary contribution is a cost function that evaluates the agent's position within the scenario, with the optimization of this cost function yielding the optimal positions for the agents.

**Strengths:**

The paper is easy to read and the proposed method is simple and straightforward.
The addition of figures is useful for comprehending the approach and the setting. The related work seems complete and correctly frames the work in the current literature.

The cost function and its technical details are well presented and the related equations are sound.

Moving on to the experiments the proposed method is mainly compared to POSA and professionals who were asked to place the agents manually. The comparison to the professionals is very interesting and the proposed method returns results that are competitive.

Overall, the introduction of the novel cost function is interesting for the task and returns good results in the presented experiments.

**Limitations:**

The main limitations of this paper are related to the experimental evaluation of the proposed cost function. While the results could be interesting and comparable to those achieved by professionals manually placing the agents, the only baseline method used for comparison is POSA, and the sole evaluated metric is a potentially biased user rating. In terms of quantitative experiments, the author shows the time needed to process a scene for each method. However, a more understandable metric for evaluating task performance should be included, perhaps even using the cost function as a metric.

Additionally, the components of the cost function and their contributions to the final performance could be evaluated through an ablation study to demonstrate that each element of the cost function is functional for the final task.

Lastly, I have concerns regarding the suitability of the paper for the multimodal community, as it lacks a clear multimodal aspect.

**Suitability:**

1

---

### Official Review · Reviewer_2FRs · 2024-05-25

**Rating:** 5
**Confidence:** 2

**Summary:**

This paper proposed a new pipeline to relocate virtual agents based on their personal characteristics in a new scenario. The features of proposed method were extracting the agent’s characteristics such as not only body shape and posture but also social distance based on past performance and preferences, and optimizing the positioning of agents in new scenes. The effectiveness of the proposed method was confirmed by comparing the positioning relationships through simulations for three virtual agents and by performing a usability testing with 30 participants.

**Strengths:**

The motivation for this study and the problems of existing studies were carefully described.
Although the proposed method was a combination of existing methods, the authors has carefully considered to establish the method.
This paper performed not only a simulation-based evaluation but also a large-scale user evaluation.

**Limitations:**

Although the effectiveness of this method could be confirmed based on the evaluation experiments, it was difficult to imagine the differences in positional relationships between proposed method reflecting individual characteristics and compared method. It would be better if an example could be shown in figures.

**Suitability:**

2

---

### Official Review · Reviewer_nz1d · 2024-05-25

**Rating:** 4
**Confidence:** 1

**Summary:**

This work introduces a system for positioning virtual agents in new scenarios based on their personal characteristics, using feature extraction and Markov Chain Mento Carlo optimization to find the optimal position.

**Strengths:**

1. The motivation is reasonable.
2. Perform qualitative and quantitative analysis.
3. The proposed method is much more efficient than the previous works.

**Limitations:**

1. The current historical modeled characteristics are limited and might not represent the human characteristics well. Developing a system that can extract various types of context to model human behavior might help.
2. The scale of the user study can be improved.

Considering the performance and efficiency of the proposed method, I think this work is borderline acceptable although some weaknesses occur.

**Suitability:**

2

---

### Meta-Review · Area_Chair_wFmV · 2024-07-01

**Recommendation:** Accept (Oral)
**Confidence:** 4

**Metareview:**

The paper presents a new system for positioning virtual agents in different scenarios based on their characteristics. The method uses feature extraction and Markov Chain Monte Carlo optimization to find the best positions for the agents. The study includes qualitative and quantitative analyses, demonstrating the efficiency and effectiveness of the proposed method compared to existing techniques.

The paper has received mixed feedback from the reviewers but is leaning towards acceptance as 4 out of 3 reviewers rated it with weak acceptance.

Reviewer nz1d: praises the motivation and efficiency of the work but noted limitations in the characteristic modeling and the scale of the user study. Reviewer 2FRs: highlights the thorough problem description and balanced evaluation despite the combination of existing methods. Reviewer 8jV3: describes concerns about the suitability for the multimodal community and the need for a more robust experimental evaluation.
Reviewer SJiL: appreciates the innovative methodology, clear writing, and practical contributions.

While the paper is well written and the approach is innovative, it is important to focus on areas for improvement by addressing current limitations highlighted by the reviewers.

Several common limitations have been identified:
1) Historical Modeled Characteristics: The current approach may not fully represent human characteristics. Improvements in context extraction could enhance the modeling of human behavior.
2) Experimental Evaluation: Concerns have been raised about the limited baseline comparison (only with POSA) and the potential bias in user ratings.
3) Suitability for the MM Community: There is a minor concern about the paper's fit within the multimodal research community.

Based on the reviews, the paper is recommended for weak acceptance. The innovative approach and comprehensive evaluations are significant strengths, while the concerns regarding experimental robustness and multimodal suitability warrant attention. Addressing these concerns in future work could strengthen the paper's contributions and applicability to the broader MM community.